# Graph Trilateration for Indoor Localization in Sparsely Distributed Edge Computing Devices in Complex Environments Using Bluetooth Technology

**DOI:** 10.3390/s23239517

**Published:** 2023-11-30

**Authors:** Yashar Kiarashi, Soheil Saghafi, Barun Das, Chaitra Hegde, Venkata Siva Krishna Madala, ArjunSinh Nakum, Ratan Singh, Robert Tweedy, Matthew Doiron, Amy D. Rodriguez, Allan I. Levey, Gari D. Clifford, Hyeokhyen Kwon

**Affiliations:** 1Department of Biomedical Informatics, School of Medicine, Emory University, Atlanta, GA 30322, USA; yash@dbmi.emory.edu (Y.K.); soheil.saghafi@dbmi.emory.edu (S.S.); hyeokhyen.kwon@emory.edu (H.K.); 2School of Electrical and Computer Engineering, Georgia Institute of Technology, Atlanta, GA 30332, USA; 3Department of Neurology, School of Medicine, Emory University, Atlanta, GA 30322, USAamy.rodriguez@emory.edu (A.D.R.); alevey@emory.edu (A.I.L.); 4Department of Biomedical Engineering, Georgia Institute of Technology and Emory University, Atlanta, GA 30322, USA

**Keywords:** ambient health monitoring, Bluetooth low energy, indoor localization, edge computing, cloud computing

## Abstract

Spatial navigation patterns in indoor space usage can reveal important cues about the cognitive health of participants. In this work, we present a low-cost, scalable, open-source edge computing system using Bluetooth low energy (BLE) beacons for tracking indoor movements in a large, 1700 m${}^{2}$ facility used to carry out therapeutic activities for participants with mild cognitive impairment (MCI). The facility is instrumented with 39 edge computing systems, along with an on-premise fog server. The participants carry a BLE beacon, in which BLE signals are received and analyzed by the edge computing systems. Edge computing systems are sparsely distributed in the wide, complex indoor space, challenging the standard trilateration technique for localizing subjects, which assumes a dense installation of BLE beacons. We propose a graph trilateration approach that considers the temporal density of hits from the BLE beacon to surrounding edge devices to handle the inconsistent coverage of edge devices. This proposed method helps us tackle the varying signal strength, which leads to intermittent detection of beacons. The proposed method can pinpoint the positions of multiple participants with an average error of 4.4 m and over 85% accuracy in region-level localization across the entire study area. Our experimental results, evaluated in a clinical environment, suggest that an ordinary medical facility can be transformed into a smart space that enables automatic assessment of individuals’ movements, which may reflect health status or response to treatment.

## 1. Introduction

Over the past decade, the presence of smart connected devices has led to a new wave of ambient monitoring of patients in clinical environments. Patients’ movements in indoor spaces play a vital role in assessing health, particularly cognitive health [1]. A range of solutions has been proposed to support tracking a patient’s location using various sensors, such as radio-frequency identification (RFID) [2], infrared (IR) [3], WiFi [4], and Bluetooth [5] signals. Among them, Bluetooth low energy-based (BLE-based) systems have gained popularity for their unique advantages, offering a low-cost, low-power, and privacy-preserving solution. Despite their many successes, BLE-based localization systems require a dense, uniform distribution of BLE receivers to track detailed movements in space. Even then, they struggle in noisy environments [6,7].

For large indoor spaces with complex structures, like hospitals, BLE solutions have been primarily used as a room-level localization technique, limiting the understanding of detailed behaviors of individuals [8]. Conventionally, the received signal strength indicator (RSSI) is used for BLE-based localization as it is a function of the distance between the receiver and the BLE beacons/transmitters [9]. However, in wide indoor spaces with multiple regions and furniture of varying materials, RSSI can be inconsistent and unstable across the area. This is due to environmental factors, such as absorption, diffraction, reflection, and interference. This inconsistency and instability significantly hamper the localization accuracy [10].

In this work, we introduce a novel indoor localization system that can accurately localize multiple subjects while they navigate in a space of 1700 m${}^{2}$ in area, designed to carry out therapeutic services for participants with Mild Cognitive Impairment (MCI). MCI is a clinically recognized stage between normal aging and dementia, which is marked by a decline in cognitive functions like memory and attention. A key feature of our system is its ability to perform reliably, even in environments with sparse distribution of edge devices. This irregularity is further exacerbated by architectural features, yet our system is engineered to hedge against such challenges. Within the study space, participants with varying degrees of MCI engaged in various therapeutic activities, including physical exercise and memory training, while fostering social interaction opportunities. Automated patient localization and tracking during diverse activities can provide important clues about cognitive health.

The proposed BLE-based localization method, graph trilateration, is developed with sparsely distributed edge computing systems that use the temporal information of RSSI received from the beacons over time. Graph trilateration effectively tackles the non-uniform RSSI coverage resulting from the aforementioned environmental factors, coupled with a sparse distributions of BLE receivers in the study space, and provides precise and robust localization solutions in noisy environments. The proposed privacy-preserving framework is also cost-effective and easy to install, owing to the low cost of edge computing devices (approximately $200 per device). Our system is versatile and can be applied to analyze the movement and interactions of participants in various contexts, such as healthcare facilities, where this study was conducted. We expect the proposed framework to transform typical indoor spaces into smart environments, providing low-cost, opportunistic services for ambient behavior assessments.

## 2. Related Work

Indoor localization using BLE is an active area of research and is deployed largely in two different approaches, *fingerprinting* [11,12,13,14,15,16,17] and *trilateration* [18,19,20,21,22]. Fingerprinting is known for its simplicity with bounded positioning error. This method first captures the radio frequency patterns at regularly spaced locations within an area, which are subsequently used as fingerprints of radio frequencies. The BLE beacons carried by individuals are localized by matching their detected patterns from a database of radio frequency fingerprints. The error bound is determined by the resolution of the reference radio frequency grid map collected in the database. Hence, it is difficult to scale this method as the size of the indoor space increases. Also, BLE beacons are required to be stationed in fixed regions and any slight alteration in the beacon’s position can lead to mismatches and inaccuracies, requiring recalibrating and collecting radio frequency fingerprints in the spaces [23,24]. In our study site (https://empowerment.emory.edu/how-we-help/environment.html accessed on 16 November 2023), our edge devices had to be replaced or repositioned frequently making the fingerprinting method infeasible for a BLE-based localization system.

Trilateration determines individuals’ positions by calculating the relative distance from known locations of multiple BLE beacons. To find the location accurately, RSSI values must be stable and at least three beacons need to be detected. This is challenging in real-world conditions, as many real-world indoor facilities have complex interiors, which include metallic structures and furniture. In such an environment, BLE signals become absorbed, diffracted, or interfered with, leading to severe degradation in localization accuracy [7,25,26]. This hampers the application of BLE-based localization techniques to monitor subject navigation in complex indoor spaces like hospitals. Our study site also had significant environmental challenges. We had edge devices that were located in the ceiling, surrounded by intricate metallic structures, which interfered with the Bluetooth signal paths, resulting in fluctuating RSSI values as subjects navigated around edge devices. Furthermore, BLE beacons have a limited range of signal propagation. This requires dense installation of BLE beacons across the facility to ensure that a minimum of three beacons are detected at any location. However, such a dense installation of BLE beacons requires specialized instrumentation and modification of existing infrastructure, which is costly and inapplicable in resource-limited environments.

In response to the complexities presented by standard trilateration, the proposed work introduces a graph trilateration approach applicable to complex, vast indoor spaces, even when RSSI signals are inconsistent and noisy and BLE beacons are sparsely distributed. We tackle the challenges by incorporating the temporal context of RSSI with the graph-based technique.

## 3. Method

### 3.1. Indoor Localization Using Graph Trilateration

We proposed a novel BLE-based indoor localization technique that adapted a trilateration method using RSSI signals from multiple BLE beacons. The RSSI, measured in decibels (dB), reflects the received signal strength and is influenced by the Free Space Path Loss (FSPL), which quantifies the loss of signal strength as an electromagnetic wave propagates through free space. Our beacons broadcast at a power of −12 dBm, and the distance to an edge device, *d*, is calculated from the RSSI values according to the FSPL model [27]:(1)$$FSPL\;\left(\mathrm{dB}\right)=20\log_{10}\left(d\right)+20\log_{10}\left(f\right)+20\log_{10}\left(\frac{4\pi }{c}\right)$$

The FSPL can be adapted to account for indoor environments by incorporating the path loss exponent *N*, which adjusts for the environmental impact on signal propagation. Thus, the distance *d* can be estimated using the modified FSPL equation [28]:(2)$$d=10^{\frac{M_{RSSI}-I_{RSSI}}{10N}}$$
where $M_{RSSI}$ is the RSSI at the reference distance of 1 m from the beacon, and $I_{RSSI}$ is the instantaneous RSSI measured by the receiver. *f* represents the frequency of the BLE signal, and *c* is the speed of light. We should note that variations in the path loss exponent *N* due to obstacles (e.g., complex ceiling structures), human movement, and multipath propagation could cause modeling errors here.

*N* represents environmental surroundings near and between the receiver and transmitter, which is not only affected by the medium for electromagnetic wave propagation, such as air ducts or metal structures in the ceiling where edge computing devices are located but also is affected by the angular position between the BLE beacon and the stationary transmitter on the ceiling. We determined the value of *N* through manual measurements of $M_{RSSI}$ for each edge computing device, taking into account there is a different metallic structures surrounding each device in the ceiling. The study determined that the range of variation for *N* spanned from 2 to 4, with the consistently optimal value being 3.5, leading to its choice as a fixed value for all cases.

Due to the sparse distribution of edge computing devices, it is highly likely that less than three edge computing devices are detected from a BLE beacon at the same time while the subject navigates the space, which prevents the application of standard trilateration. Our graph trilateration integrates a sliding window approach with a graph-inspired technique to tackle noisy environments with sparse BLE receiver distributions. Due to the structural complexities, RSSI is not consistent around the edge device, and RSSI can be very small even when a subject is close to the edge computing device, which can significantly distort the localization outcome with standard trilateration. We tackle this challenge by additionally using the received hits strength index (RHSI), which considers the number of hits (or detections) made by edge computing devices independently from the RSSI within a given time window *t* s, while a subject navigates in the space. Furthermore, within a certain time window (time window (a.k.a window size (or τ) in the following section) is a hyperparameter and could not be less than 0.5 s as the BLE beacons’ frequency is 2 Hz), we could receive multiple hits from a particular edge device. As mentioned earlier, relying solely on the RSSI value for determining the subject’s location can lead to inaccuracies. Primarily this problem arises because, even when the subject is near the edge computing device, the RSSI value might be low due to a complex ceiling structure. However, if we include the count of hits from the computing edge device in the algorithm, it has the potential to greatly enhance the accuracy of the localization process. In the following section, we are going to investigate this idea by solving the trilateration problem as a graph-based approach, considering all edge devices detecting a BLE beacon as nodes, $\left\{p_{1},p_{2},\cdots ,p_{M}\right\}$, and their pair-wise connectivity as edges, $p_{ij}$, where 1≤i,j≤M and i≠j, when *M* edge devices detect BLE beacon.

The graph trilateration technique takes a two-step approach. It first considers edge-based BLE beacon locations derived from all pairs of edge devices and then derives the final location of the BLE beacon by aggregating edge-based locations. We propose two different approaches for graph trilateration depending on how RHSI is used, either by aggregating the estimation of pairwise edge-base BLE beacon locations with the RHSI-based weighted sum, *RHSI-Agg*, or by incorporating the RHSI-based weights into the pairwise edge-based BLE beacon localization and computing the final location by taking the average, *RHSI-Edge*. In the graph trilateration methodology, using either *RHSI-Agg* or *RHSI-Edge*, the estimated location is further refined by incorporating the previous estimation with defined weights, known as a smoothing artifact. In other words, the BLE beacon location detected at consecutive *T* windows, $\left\{L_{t},L_{t+1},\cdots ,L_{t+T}\right\}$ is further smoothed in time with temporal weights for temporarily consistent spatial navigation of detected BLE beacons.

#### 3.1.1. Node Property

Each edge computing device of the graph, $p_{i}=\left(h_{i},d_{i}\right)$, is defined as RHSI, $h_{i}$, and the distance, $d_{i}$, between the detected beacon and edge computing device, $p_{i}$ over the *t* sec time window. Due to the fluctuations and inconsistency in RSSI, the number of RSSI hits, $h_{i}$, within *t* sec varies according to the subject’s navigation path and we consider the representative RSSI of $p_{i}$ as the average of RSSIs observed in the time window, $RSSI_{i}=\frac{1}{h_{i}}\sum_{k=1}^{h_{i}}RSSI_{k}$, where $\left\{RSSI_{1},\cdots ,RSSI_{h_{i}}\right\}$ is detected within *t* s. For each $p_{i}$, $d_{i}$ is derived from $RSSI_{i}$ using Equation (Equation 2).

#### 3.1.2. RHSI-Agg Trilateration

We first geometrically derive the lateral distance, $r_{i}$, between the detected beacon and edge computing device, $p_{i}$, from $d_{i}$ and the height of the ceiling, measured from the waist (approximated as 2 m), as shown in Figure 1A. Then, the potential location of the BLE beacon, $m_{i,j}$, between every edge from edge device pairs, $p_{i,j}$, using a weighted average of lateral distance, $r_{i}$ from each edge computing device is given by:(3)$$m_{i,j}=\left(x_{i}+\frac{r_{i}}{r_{i}+r_{j}}\left(x_{j}-x_{i}\right),\;y_{i}+\frac{r_{i}}{r_{i}+r_{j}}\left(y_{j}-y_{i}\right)\right)$$
where $p_{i}=\left(x_{i},y_{i}\right)$ and $p_{j}=\left(x_{j},y_{j}\right)$ are 2D coordinate locations of the top-down view of our study site. The locations of BLE beacons are considered as the weighted average of the overall RHSI observed from $p_{i,j}$, $w_{i,j}=h_{i}+h_{j}$, since a higher RHSI indicates a higher likelihood of the BLE beacon being closer to an edge device. The final estimated location of the BLE beacon is $L_{RHSI-Agg}=\frac{\sum_{i}^{n}\sum_{j}^{n}w_{i,j}m_{i,j}}{\sum_{i}^{n}\sum_{j}^{n}w_{i,j}}$.

#### 3.1.3. RHSI-Edge Trilateration

Here, we use RHSI when deriving edge-based BLE beacon location, $m_{i,j}$, to consider RSSI fluctuations for each $p_{i}$ separately. For an edge device pair, $p_{i,j}$, $d_{i}$ and $d_{j}$ are first weighted with the ratio of the RHSIs for each device pair, $W_{i,j}=\frac{h_{i}}{h_{i}+h_{j}}$ and $W_{j,i}=\frac{h_{j}}{h_{i}+h_{j}}$, as shown in Figure 1B. Then, the weighted lateral distances, $r_{i}$ and $r_{j}$, are geometrically derived as $W_{i,j}d_{i}$ and $W_{j,i}d_{j}$, with the height of the ceiling from a subject’s waist approximated as 2m. The potential location of the BLE beacon, $m_{i,j}$, between every edge from edge device pairs, $p_{i,j}$, is estimated similarly to RHSI-Agg trilateration, $m_{i,j}=\left(x_{i}+\frac{r_{i}}{r_{i}+r_{j}}\left(x_{j}-x_{i}\right),\;y_{i}+\frac{r_{i}}{r_{i}+r_{j}}\left(y_{j}-y_{i}\right)\right)$. Finally, the location of the BLE beacon is derived by averaging all $m_{i,j}$:$$\begin{array}{c} L_{RHSI-Edge}=\frac{1}{N(N-1)}\sum_{i\neq j}m_{i,j}. \end{array}$$

#### 3.1.4. Handling Corner Cases

The proposed method uses a sliding window approach to aggregate RSSI and RHSI over time so that at least three edge computing devices detect BLE hits to apply graph trilateration. Yet, due to the sparse distribution of edge computing devices and noisy environments common in real-world conditions, corner cases arise where fewer than three edge devices detect hits in a given *t* sec window. We handle such cases as follows:Within a given time window, *t*, when only a single edge computing device is detected, we assume the BLE beacon location to be directly beneath the edge computing device, regardless of RSSI, $L^{t}=p_{i}^{t}={\left(x_{i},y_{i}\right)}^{t}$.When two edge computing devices are detected, we approximate the BLE beacon location as the edge-based localization, $L^{t}=m_{i,j}^{t}$ for $p_{i,j}$ from either *RHSI-Agg* or *RHSI-Edge*, accordingly.

In rare cases, consecutive *T* time windows, {t,t+1,⋯,t+T}, may contain the same single or two edge computing devices detected, which is possible where power and network sources are extremely sparse in the region to install edge computing devices. In such cases, we temporally interpolate the locations between the first, $L^{t}$, and the last, $L^{t+T}$, timestamps, $L^{t+k}=\frac{kL^{t}+\left(T-k\right)L^{t+T}}{T}$.

#### 3.1.5. Enhancing Temporal Consistency

While incorporating additional information from prior estimated locations has negligible effects when their associated temporal weights are small, assigning higher temporal weights to these previous estimates will significantly enhance the correlation between the next prediction and its predecessors. In terms of not facing this issue, we ended up using the previous estimated location information at most three times (i.e., T=3). More specifically, for this T=3 consecutive locations, $L^{t:t+T}=\left\{L_{t},L_{t+1},\cdots ,L_{t+T}\right\}$ detected either from *RHSI-Agg* or *RHSI-Edge* trilateration along with temporal interpolation for corner cases, we further apply weighted smoothing for temporal consistency in BLE beacon locations, as it is unlikely for subjects to navigate in a zig-zag manner. The localization at time t+T is estimated as $L^{t+T}=S\cdot L^{t:t+T}$, where $S=\{s_{1},s_{2},\cdots ,s_{T}\}$ such that $s_{1}<s_{2}<\cdots ,<s_{T}$, as illustrated in Figure 2C.

## 4. Benchmark Data Collection

The therapeutic space consists of various regions, including a gym, kitchen, library, dining area, open theatre, maker lab, and other bespoke regions designed to facilitate therapeutic activities for cognition for participants with MCI (see Figure 3A). To track movements in a wide space with a complex structure, we designed an edge computing framework that comprises 39 edge computing units, each containing a Raspberry Pi 4 model B (4GB of RAM) equipped with a BLE 4 antenna that is placed on the ceiling [29,30], shown in Figure 3B, where network and power sources are accessible nearby. The edge computing devices were placed non-uniformly and sparsely throughout the space to work with the existing network and power sources of the infrastructure, thereby avoiding any modifications to the existing infrastructure. Our subjects were equipped with a belt bag containing a BLE beacon (Smart Beacon SB18-3 by Kontact.io). As the subjects move around, edge computing devices within proximity record the received signal strength indicator (RSSI) transmitted from the BLE Beacon. BLE beacons are scanned at a sampling frequency of 2 Hz using the Bleak [31] Python package. This RSSI data were subsequently relayed to an on-premise server, compliant with the Health Insurance Portability and Accountability Act (HIPAA). This was done in real-time through the wired network. The on-premise server analyzed the collected RSSI from all edge computing devices to localize subjects using the proposed method. Our centralized approach for indoor localization and data management allowed a secure system to prevent data breaches or leaks of sensitive data.

Our benchmark dataset was gathered with the assistance of three young and healthy participants (aged 26.3±4.02 years, with an average height of 173.3±5.9 cm) following specified paths in the study site, as depicted in Figure 4. To improve the accuracy of our ground truth data throughout the data collection process, we have placed markers (A physical or visual indicators that serve as a reference point for measurements or observations, allowing for precise and consistent data collection) at one-meter intervals throughout the entire space in our facility where most of our therapeutic activities take place.

An observer identified and recorded the nearest marker to each participant as they moved. The subjects were asked to move within no later than 10 s to simulate realistic spatial navigation that normally occurs in our therapeutic facility. This procedure yielded a dataset that comprehensively covers our therapeutic spaces comprising 105 data points for each participant, allocated as follows: 28 locations in the right corridor, 28 locations in the left corridor, 21 locations in the kitchen, 8 locations in the lounge, and 20 locations in the activity area. These regions are highlighted in Figure 3A.

## 5. Evaluation Metrics

### 5.1. Evaluating Multi-Person Localization

We evaluated the performances of the proposed method on two levels: positioning-level and region-level localization. The positioning error was calculated as Euclidean distance in meters between the estimated and ground-truth locations to understand the error margin of tracking fine-grained movements. The region-level localization was computed as the percentage of times each region was predicted correctly while the subject moved within the regions specified in Figure 4A–C. As much as granular movements are essential, understanding region-dwelling profiles can help understand the routine activities of subjects while participating in therapeutic activities. To assess the generalizability of our proposed methods, RHSI-Agg and RHSI-Edge, we employed a leave-two-subject-out cross-validation approach. This allowed us to fine-tune parameters that were traditionally kept constant in the localization algorithm, as illustrated in Figure 2. In the process of fine-tuning and optimization, we employed the grid search method to minimize the mean square error using a leave-two-subject-out cross-validation approach. The average positioning error and room-level accuracy, are summarized in Table 1 and Table 2, along with their corresponding standard deviations. These hyperparameters are listed below:**Window Size** (τ): The size of the time window considered for localization, varying between 0.5 to 60 s (Figure 2A).**Slide/Step**: Interval of sliding window in time (Figure 2B). For the sliding method, we used a 1-second sliding interval to ensure overlaps in sliding windows as short as τ = 2 sec. For the step method, we used the same size of the sliding interval with τ to avoid overlaps in windows. The sliding method provides more temporally smoothed BLE localization results due to overlapping temporal context in subsequent sliding windows.**Weighting Factors (*****S*****; Section 3.1.5)**: Temporal smoothing weights for *T* consecutive localizations (Figure 2C). We explored seven weighting factors (i.e., $S_{1},S_{2},\cdots ,S_{7}$), where each weighting factor is a vector with three elements (e.g., $S_{1}=\left[s_{1_{1}},s_{1_{2}},s_{1_{3}}\right]$) for T=3, indicating the degree of dependency from past locations. We set $S_{i}$ to have more dependency on the temporally further $L_{t}$ with increasing i=1,⋯,7 ($S_{1}=\left[0.0,\;0.0,\;1.0\right],\;S_{2}=\left[0.0,\;0.1,\;0.9\right],\;S_{3}=\left[0.0,\;0.2,\;0.8\right],\;S_{4}=\left[0.0,\;0.3,\;0.7\right],\;S_{5}$$=\left[0.0,\;0.4,\;0.6\right],\;S_{6}=\left[0.1,\;0.2,\;0.7\right],\;S_{7}=\left[0.1,\;0.3,\;0.6\right]$).

### 5.2. Baseline Method: Standard Trilateration

We compare the proposed method with the standard trilateration method [32], which estimates the location of the BLE beacon at a given timestamp at the intersection of circles with radius $r_{i}$ centered at each edge device, $p_{i}$. To identify the exact location of the BLE beacon, at least three edge devices need to detect RSSI from the BLE beacon. The accuracy of the estimated location increases as more edge device detects the BLE beacon. Considering the sparse distribution of edge computing devices in our study area, we also apply a sliding window approach to the locations detected by standard trilateration, $L_{std}^{t}$, at time *t*, for a fair comparison with the proposed method. The hyperparameters are optimized using the above-mentioned cross-validation approach.

## 6. Results

Table 1 shows the overall positioning errors with standard deviation across the entire study site using our proposed method compared to the baseline method. Standard trilateration is evaluated with a sliding or step window approach. Graph trilateration method is compared with and without interpolation, with *RHSI-Agg* or *RHSI-Edge*, and with a sliding or a step window approach. Overall, the lowest positioning error was 4.44 m when graph trilateration with RHSI-Edge was used without interpolation, which was 2 m lower than when using the standard trilateration method with sliding windows. On average, across varying graph trilateration methods, *RHSI-Agg* showed 0.03 m lower error than *RHSI-Edge*, the step window-based approach showed 0.5 m lower error than the sliding window-based approach, and applying temporal interpolation across corner cases showed 0.3 m higher errors compared to the methods without interpolations.

Table 2 shows the positioning error and room-level localization accuracy in different regions for the best methods using standard trilateration and graph trilateration from Table 1. We also show the number of edge computing devices and the size of each region for signal coverage. Positioning error and room level accuracies varied across the regions in our facility. The proposed graph trilateration method significantly outperformed standard trilateration across all regions. The region-by-region performance also varied for graph trilateration, with and without temporal interpolation. For the right corridor and lounge regions, temporal interpolation for corner cases decreased the positioning error to 0.68 m on average, whereas for other regions, the positioning error increased to 0.67 m. For room-level localization, introducing temporal interpolation resulted in an overall improvement of the model performance by 1.08%.

## 7. Discussion

### 7.1. Indoor Localization Performance

We observed a significant improvement in performance using graph trilateration instead of standard trilateration (resulting in a 2 m decrease in positioning error). Our work demonstrates that RHSI is useful in noisy environments where RSSI is inconsistent because it also considers weak RSSIs (potentially due to the noisy environment) when subjects are navigating near edge computing devices. We hypothesize that this is because standard trilateration considers all edge computing devices at the same time to find a global solution for the BLE beacon location, which can be distorted if a subset of recorded RSSIs does not represent the distance of the beacon accurately. On the other hand, the graph trilateration method takes a two-step approach to first find local solutions between pairs of edge computing devices, which is aggregated afterward. This shows that it is important to provide an adaptive approach to consider the contribution of each edge computing device, as they interact non-linearly with BLE beacons in the noisy environment. Also, the *RHSI-Edge* considers pair-wise RSSI for every beacon that is additionally integrated with RHSI, and *RHSI-Agg* only uses RHSI when aggregating RSSI-based pairwise localizations, $m_{i,j}$. Both approaches were comparable but with a slight decrease in positioning error (0.03 m) for *RHSI-Agg*, which demonstrates that considering RHSI is more useful when taking into account every edge computing device involved for BLE beacon localization once pairwise local solutions have been derived.

We also analyzed the impact of the density of the edge devices with regard to localization performance. From Table 2, we define the density of the edge devices at each region, $\rho =\frac{\#}{m^{2}}$, as the number of edge devices, #, per unit region size ($m^{2}$).The device density and positioning error showed a negative correlation (with a slope of −0.33). This demonstrates the significance of challenges faced when sparse coverage of edge computing devices is used for BLE-based localization. Yet, our results could localize BLE beacons as low as 2.51 m positioning error (Left Corridor) and over 90% room-level localization accuracy (Right Corridor, Left Corridor, and Activity Area). This average positioning error represented in Table 2 serves as indicative metrics that inherently capture the system’s adaptability to varying indoor conditions.

### 7.2. Impact of Hyperparameters

Interestingly, temporal interpolation applied to handle corner cases showed mixed results. The positioning error slightly increased in the left corridor, kitchen, and activity area and decreased in the right corridor and lounge as show in Table 3.

Additionally, the room-level accuracy improved for the lounge and activity regions, while, it decreased in the left and right corridor and the kitchen. From our observations, the errors came from scenarios where the closest edge computing devices are located in the boundary of signal ranges (approximately 10 m) of BLE beacons. As the subject moves, the BLE beacon is randomly detected by multiple edge computing devices rapidly switching between one another, due to a significantly unstable RSSI. Figure 5A,B illustrate two cases when not using interpolations. In Figure 5A, the nearest edge computing device from the beacon registers four hits, while the device located further has only one hit. On the other hand, in Figure 5B, the closer edge computing device has one hit, while the device further away has four hits. Figure 5C shows the estimated location from temporal interpolation, which smooths out the random detections by the edge computing devices to provide a balanced solution between Figure 5A,B. When assuming vertical room boundaries in the middle of two edge computing devices, Figure 5A has the lowest positioning error and highest room-level localization accuracy, while Figure 5B has the highest positioning error and lowest room-level localization accuracy. Consequently, Figure 5C has the positioning error and room-level accuracy in between those from Figure 5A,B.

Interpolation relies on the strong but often unrealistic assumption of linear motion, which can result in a substantial error dependency on both the positions of edge devices and the movements of individuals. Our results in the following section indicate that the non-interpolated method yields results that are nearly identical to the interpolated one. This suggests that the underlying linear assumption lacks robust quantitative support.

Between the slide and step window methods, the sliding window showed worse localization error increasing by 0.5 m on average. We observed that sliding windows with overlaps derived significantly smoothed artifacts, missing the important location transition in between.

The window size was also an important factor to consider especially when deploying a sparsely distributed edge computing system. The trilateration-based approach requires having at least three edge computing devices to identify the unique locations of BLE beacons. Without a windowing approach, or not considering temporal context, only a single or two edge computing devices were activated for the majority of the cases. We discovered that a 40-s window was optimal for the balance between the number of cases detecting three edge computing devices, which we used in our experiments. Further increasing the window size could also increase the number of windows containing more edge computing devices, but this significantly decreased localization performance, as increasing the window size has the effect of averaging out the locations of BLE beacons over a longer time duration.

To understand the effect of the varying weighting factors, $S_{1:7}$, we explore the changes in positioning errors in the training dataset, when the window size is varied using *RHSI-Edge* trilateration experiments with slide or step method and with or without interpolation. The results are shown in Figure 6, where a lighter color (yellow) means lower positioning error, and vice versa. Figure 6A shows positioning errors from the slide method without interpolation, Figure 6B shows positioning errors from the slide method with interpolation, Figure 6C shows positioning errors from the step method without interpolation, and Figure 6D shows positioning errors from step method with interpolation. Shown in Figure 6A,B, across all hyperparameters, the slide method showed minimal variation regardless of weighting factors, $S_{1:7}$, or the application of interpolation. We consider that overlaps in the temporal context in subsequent locations induce excessive smoothing error when combined with temporal smoothing using $S_{1:7}$. The overall positioning error was also higher than the step method. For the step methods (Figure 6C,D and Table 2), weighting factors had a significant impact. Both with and without interpolation, the lowest positioning error was shown for window sizes between 40 and 50 s and weighting factors of $S_{3,4,5}$. For weighting factors, $S_{3,4,5}$, we only consider subsequent locations from the current and the previous timestep. These results show that temporally smoothing across an extended period of time can accumulate positioning errors over time. Overall, this analysis shows the importance of finding the sweet spot for the duration of the temporal context when using BLE-based localization to tackle challenging scenarios like those in our study.

### 7.3. Impact of Edge Device Distribution

In our experiments, we consistently found that in over 95% of instances, only 1–2 unique edge devices are detectable within a 5-s timeframe. This limitation prevents the effective use of the trilateration method and compromises the reliability of fingerprinting techniques for localization. This constraint is further compounded by the variable signal reception quality across the study area, as illustrated in the subsequent analysis. Figure 7 shows the heatmap representations of the signal reception across the study area, which is collected at the equally spaced 131 locations in our study site. The signal reception and strength across our study site are inconsistent, with some regions such as the Right Corridor and Lounge areas exhibiting high positioning error as shown in Table 2. RSSI can fluctuate due to environmental factors, such as metallic structures in walls or furniture, which can block or redirect the signal [33]. As expected, local regions with more edge devices, such as the left corridor, kitchen, and activity area, have higher RHSI (Figure 7A) and higher average RSSI (Figure 7B), which means more robust and more consistent signal strength from beacons. As a result, those areas demonstrated lower positioning errors in Table 2 compared to other regions. Signal coverage analysis explains the poor performance of standard trilateration in our study site, which depends on accurate distance measures between each edge devices and beacons.

### 7.4. Limitations and Future Works

Our proposed method assumes that all edge devices operate under the same environmental conditions, where the same *N* value in Equation (Equation 2) is used across all edge computing devices. But in reality, conditions vary from one location to another significantly as different regions in our facility have their own specific space designs with furniture to serve their purpose. For example, the kitchen, lounge, and activity area are designed with varying materials. We believe a potential solution is to introduce a bias factor, validated on the RSSI and RHSI received in each region as a function of the edge computing device involved in the localization. This would help adjust for different conditions and their effects on signal quality. In our study, we address the query of why longitudinal components (*z*-axis) are disregarded in our measurement system, where the ceiling-to-waist distance is fixed at 2 m for each subject. By considering the height differences with variations in leg length and body style in different subjects, their corresponding distance to the ceiling comes from the normal distribution with a mean of 2 m and a few centimeters difference in its first standard deviation. A few centimeters difference could be negligible after mapping the estimated location in the lateral distance.

While our graph trilateration approach can track spatial navigation using sparsely distributed edge computing devices with an error bound of 4.44 m on average, it is insufficient for capturing small and more nuanced fine-grained movements while navigating our facility, which would provide useful movement information to understand the cognitive impairments of our subjects. The primary cause of this error comes from the implementation of a larger window size, employed to compensate for the limited coverage of edge devices within smaller time frames. To obtain a more detailed view, we are considering combining our current method with data from inertial measurement unit (IMU) sensors. Previous work demonstrated a significant improvement in navigation tracking using BLE and IMU sensor fusions [34,35,36,37].

Through this multimodal fusion, we hope to obtain a more precise picture of how individuals move within indoor spaces.

Nevertheless, with our current algorithm, we can detect room-level localization with an average of 85% accuracy. Monitoring room-dwelling behavior can serve to detect abnormal activity in our subjects with MCI while taking therapeutic training in our facility, such as leaving the classroom in the middle of the class or visiting the restroom more frequently than expected. We will study the validity of our algorithm for assessing behaviors related to MCI in future works.

## 8. Conclusions

In this paper, we propose an open-source, scalable indoor localization approach using BLE sensors and sparsely distributed edge computing systems for extensive indoor regions. Our analysis reveals that in large indoor spaces (over 1700 m${}^{2}$) with intricate structures, an uneven distribution of edge devices can lead to inconsistent signal coverage, which poses significant challenges for BLE-based localization approaches, especially with the standard trilateration method which inherently assumes a dense and stable RSSI coverage. Our proposed graph trilateration method leverages the temporal density of hits from BLE beacons, namely RHSI, integrated into a graph-based approach, which can pinpoint subjects’ positions with an average error of 4.4 m across the entire study area. Additionally, it achieves over 85% accuracy rate for region-level localization. In our future research, we plan to deploy the proposed system to study spatial navigation behaviors in subjects with MCI, which is known to provide biomarkers for cognitive impairment. We expect that the proposed graph trilateration localization technique will help medical practitioners transform any therapeutic facility into a smart space that can passively monitor patients’ behaviors to provide evidence-based care tailored to individual patient conditions [1,38].

## Figures and Tables

**Figure 1 sensors-23-09517-f001:**
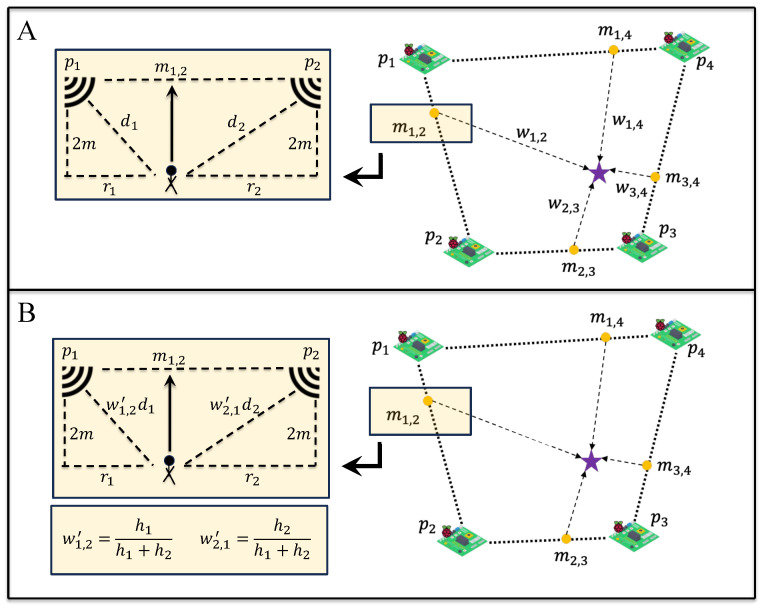
Graph trilateration approach. (**A**) **RHSI-Agg** trilateration method, first calculates potential locations, $m_{i,j}$, between each $p_{i}$s (i.e., $p_{i}$ and $p_{j}$). Then $m_{i,j}$s are aggregated with weights ($w_{i,j}=h_{i}+h_{j}$) that computed based on the RHSIs, $h_{i}$ and $h_{j}$, where i,j=1,2,3,4, to localize the subject (purple star in this illustration). (**B**) **RHSI-Edge** trilateration method first incorporates RHSIs, $h_{i}$, to calculate the distance between the subject and the positions of the $p_{i}$s. Then, it computes the average of all estimated edge-based locations to pinpoint the subject’s location. Note: In the figure, for the sake of simplicity, we just considered i=1 and j=2 for the visualization part.

**Figure 2 sensors-23-09517-f002:**
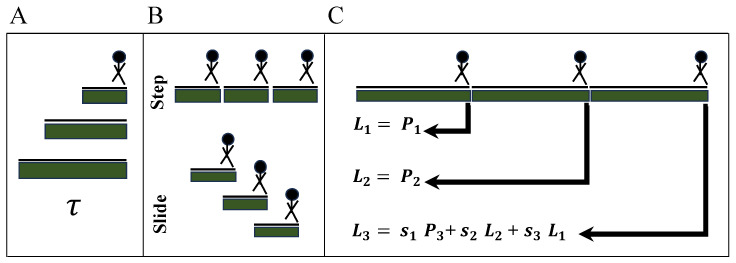
Hyperparameters used in graph trilateration. (**A**) τ represents the size of the time window to consider the temporal context in the proposed method. (**B**) The step method uses non-overlapping sliding windows, where the window size and sliding interval are the same. The sliding method indicates overlaps in sliding windows to induce more temporally smooth BLE localization. (**C**) The weighting factor applies temporal smoothing to BLE locations estimated over T=3 timesteps using the weighted average. We explore various weighting values with $s_{1}<s_{2}<s_{3}$.

**Figure 3 sensors-23-09517-f003:**
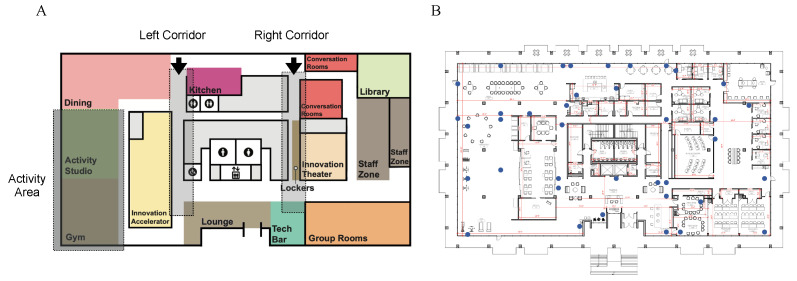
Study site and data collection. (**A**) Study sites are designed to include various utility spaces. (**B**) The locations of 39 edge computing devices (Raspberry Pi v4 model B) in the ceiling.

**Figure 4 sensors-23-09517-f004:**
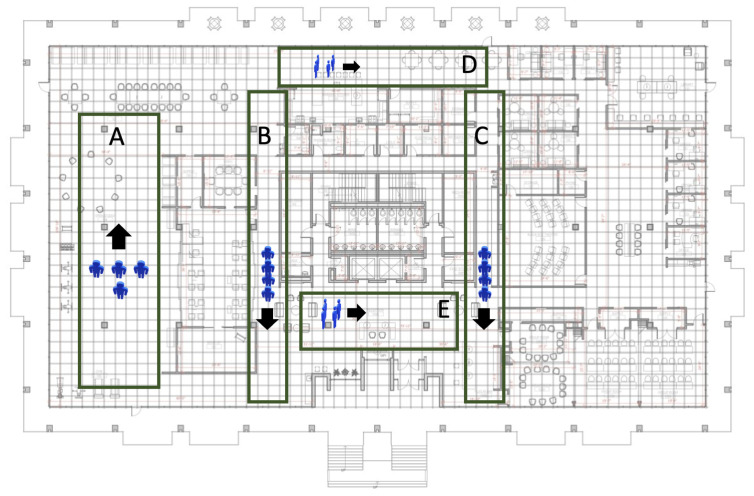
Study site and data collection—the location of individuals occupied in our data collection. (**A**) activity area, (**B**) left corridor, (**C**) right corridor, (**D**) kitchen, and (**E**) lounge.

**Figure 5 sensors-23-09517-f005:**
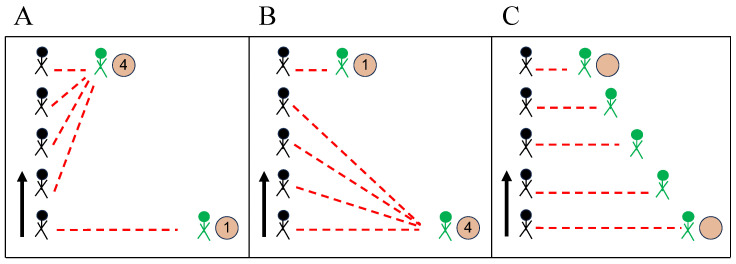
Challenges in interpolation methods. Black and green stick figures are ground truth and estimated locations of the subject, respectively, and the orange circle is edge computing devices with RHSI in numbers in a given time window. When multiple edge computing devices are in the signal boundaries from BLE beacons, as the subject moves, the BLE beacon is randomly detected by multiple edge computing devices rapidly switching between one another. (**A**) When the closest “pi” from the subject receives more hits compared to the one located further away, the error is minimized. (**B**) When the closest “pi” from the subject receives the least number of hits compared to the one located further away, the error is maximized. (**C**) The interpolation method provides a balanced solution between the two extreme cases in (**A**,**B**), which reduces variations in errors with the cost of marginally increased error compared to (**A**).

**Figure 6 sensors-23-09517-f006:**
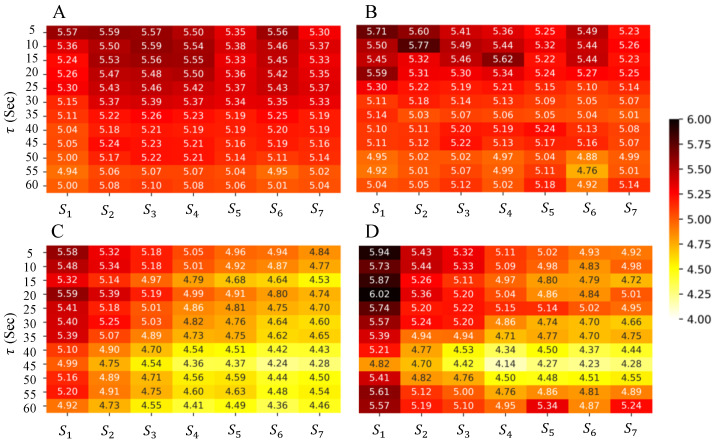
Overview of hyperparameter tuning results for *RHSI-Edge*. This figure illustrates the average training error obtained during the tuning process. (**A**) Slide—graph trilateration without interpolation, (**B**) slide—graph trilateration with interpolation, (**C**) step—graph trilateration without interpolation, and (**D**) step—graph trilateration with interpolation.

**Figure 7 sensors-23-09517-f007:**
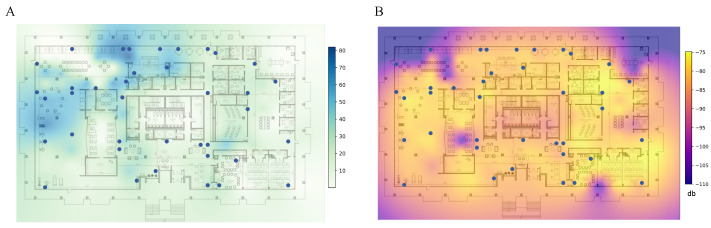
Heatmaps showing inconsistent BLE signal coverage across the study site. The (**A**) RHSI and (**B**) average RSSI signal depends on the density of edge devices and their surrounding structures.

**Table 1 sensors-23-09517-t001:** Evaluation of localization accuracy across the entire study site with different trilateration approaches.

Method	RHSI Applied	Time Window Strategy	Error ± STD (m)
Standard Trilateration	N/A	Slide	6.39±0.94
	N/A	Step	6.48±1.58
Graph (w/o Interpolation)	*RHSI-Agg*	Slide	5.01±0.41
	*RHSI-Agg*	Step	4.47±0.77
	*RHSI-Edge*	Slide	5.18±0.44
	*RHSI-Edge*	Step	4.44±0.59
Graph (with Interpolation)	*RHSI-Agg*	Slide	5.02±0.41
	*RHSI-Agg*	Step	4.60±0.26
	*RHSI-Edge*	Slide	5.04±0.15
	*RHSI-Edge*	Step	4.57±0.47

Bold values indicate the highest accuracy or the lowest error for each method.

**Table 2 sensors-23-09517-t002:** Positioning error (in meters) and room level localization accuracy (in %) in different regions using BLE.

	Right	Left			Activity	
**Method**	**Corridor**	**Corridor**	**Kitchen**	**Lounge**	**Area**	**Average**
Number of edge devices						
	6	4	5	3	7	
Region size (m${}^{2}$)						
	50	66	70	176	312	
Positioning Error (m)						
Standard Trilateration	6.43	4.11	5.99	7.27	8.18	6.39
Graph (w/o Interpolation)	4.81	**2.51**	**3.78**	4.93	**6.15**	**4.44**
Graph (with Interpolation)	**4.62**	3.91	3.99	**3.76**	6.56	4.57
Room Level Localization Accuracy (%)						
Standard Trilateration	47.98	77.06	73.97	54.39	63.98	65.74
Graph (w/o Interpolation)	**94.44**	**97.53**	**78.57**	66.66	83.33	84.11
Graph (with Interpolation)	93.82	96.29	77.77	**66.67**	**91.38**	**85.19**

Bold values indicate the highest accuracy or the lowest error for each area.

**Table 3 sensors-23-09517-t003:** Positioning error for three participants in different areas of the facility using BLE and IMU.

Signal Modality	Right Corridor	Left Corridor	Kitchen	Lounge	Activity Area	Average
BLE Positioning Error (m)	5.01	2.94	3.13	4.68	4.11	3.97
BLE and IMU Positioning Error (m)	4.31	3.43	2.57	4.71	3.21	3.65
BLE Room Level Accuracy (%)	88.1	91.4	92.7	89.2	90.2	90.3
BLE and IMU Room Level Accuracy(%)	90.7	90.6	93	90.4	91.6	91.2

## Data Availability

Data from this study are available from the corresponding author upon reasonable request.

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
