# Peer review of "Graph Trilateration for Indoor Localization in Sparsely Distributed Edge Computing Devices in Complex Environments Using Bluetooth Technology"

_sensors, 2023, doi:10.3390/s23239517_

Round 1
Reviewer 1 Report
Comments and Suggestions for Authors
The submission has nice contributions but the manuscript needs some revision to make it accurate and complete:
- The algorithmic contributions of the work are very limited with almost no novel algorithm design. The application is significant though. Hence the paper title should mention the particular application.
- The write-up needs thorough review and revision for language errors and typos.
- Accuracy of the assumed RSS model (1) should be discussed.
- Along the explanations starting on line 160, "horizontal" should be "lateral" to be more accurate. A discussion on the effect of ignoring z (longitudinal) components would be helpful.
- In the result presentation of Sections 3 and 4, there needs to be comparative analysis and discussion with respect to at least a few alternatives of the averaging based localization algorithm utilized in this draft to answer the question of why to use the localization algorithm described here instead of others from the general sensor network localization literature.
- Section 3 is too long and captures many key components, which could be presented in multiple sections. On the other hand, the simulation/ experimentation results/discussions are divided among Sections 3, 4, and 5. For clarity, a section re-organization is recommended.
Comments on the Quality of English Language- The write-up needs thorough review and revision for language errors and typos.
Reviewer 2 Report
Comments and Suggestions for Authors
The novelty of the paper topic is not very significant, however the results of the research provide the promised tools allowing to improve complex indoor navigation solutions. As the proposed method is based on the network of sensors, the paper is suitable for the Sensors scope and can be recommended for publishing. Unfortunately the quality of presentation is very low and the manuscript needs significant re-building. My recommendation is major revision and re-submission. Hope that my recommendations below will help the authors improving their manuscript correspondently.

Reviewer 3 Report
Comments and Suggestions for Authors
The work is good and in demand that is represented by the authors in "Graph Trilateration for Indoor Localization in Sparsely Distributed Edge and Cloud Computing Environments Using Bluetooth Technology" but some points of concern are there like:
1. How the work is different than "A differentially private indoor localization scheme with fusion of WiFi and bluetooth fingerprints in edge computing" and "Energy Efficient Localization through Node Mobility and Propagation Delay Prediction in Underwater Wireless Sensor Network". Is it of area only? Specify in introduction and prove the novelty and inventiveness of the work proposed.
2. I am not able to see that the performance is thoroughly evaluated based on criteria such as efficiency, precision, and adaptability across diverse technology environments.
3. Authors should present the detailed analysis of influence factors like noise levels, technology types (Wi-Fi, ZigBee, BLE), and tag node density and layout on the performance of each localization technique.
4. Is there any affect of Execution using Google Colab’s cloud-based GPU platform for optimized calibration procedures while providing insights into the proposed indoor localization technique?
5. The work should cater some good work like A new approach of location aided routing protocol using minimum bandwidth in mobile ad-hoc network
6. Overall, the work presented is good and need to take care of extension for related work.
Comments on the Quality of English Languageneed to take care of English at some points in related work.
Round 2
Reviewer 1 Report
Comments and Suggestions for Authors
Most of the pointed issues are well addressed.
Comments on the Quality of English LanguageMost of the pointed issues are well addressed.
Reviewer 2 Report
Comments and Suggestions for Authors
Dear Authors,
Reading your responses to my comments I see that you modified your manuscript significantly, thank you. However there are still some uncertainties which should be correctrd before the manuscript acceptance.
Comment#3
We updated Figure 2 (page 6) and Figure 3 (page 7) with these new notations as mentioned above....
Your comment does not correspond to the modified text of the manuscript. Please check, you probably mean Fig 1, but nof 2 and 3.
Comment#5
This allowed us to fine-tune parameters that were traditionally kept constant in the localization algorithm,as illustrated in Fig. 3.
Your comment does not correspond to the modified text of the manuscript. In the modified text I see the reference to Fig.2. Please check the text in order do not mistake with the reference.
Comment#6:
Instead of using the optimization term, we have changed our wording in the manuscript to use the term “fine-tune”.
Thank you, but "fine-tune" also means optimization or, if you prefer, residuals minimization. In turn, it means some criteria to define that your "fine-tune" really brings the minimal error in the indoor positioning. Due to this I would just recommend you to add just 1-2 sentencies to take my previous comment into account, please.
Comment#7
The main source of this systematic error (as mentioned in comment #1) for localization in the facility, arises from uneven signal distribution across large areas, resulting in sporadic edge device beacon detection.
I am very sorry, but sporadic processes can not bring stable systematic errors. Thus your current explanation sounds a bit strange. Please pay more attention to this part of your modified text to avoide contradictions.
